# Comparative Study of Cage Subsidence in Single-Level Lateral Lumbar Interbody Fusion

**DOI:** 10.3390/jcm11051374

**Published:** 2022-03-02

**Authors:** Akihiko Hiyama, Daisuke Sakai, Hiroyuki Katoh, Satoshi Nomura, Masato Sato, Masahiko Watanabe

**Affiliations:** Department Orthopaedic Surgery, Tokai University School of Medicine, 143 Shimokasuya, Isehara 259-1193, Kanagawa, Japan; daisakai@is.icc.u-tokai.ac.jp (D.S.); hero@tokai-u.jp (H.K.); nomura.s@tokai-u.jp (S.N.); sato-m@is.icc.u-tokai.ac.jp (M.S.); masahiko@is.icc.u-tokai.ac.jp (M.W.)

**Keywords:** lateral lumbar interbody fusion, cage subsidence, indirect decompression, endplate injury, lumbar degenerative disease, low back pain

## Abstract

We investigated the incidence and clinical features of cage subsidence after single-level lateral lumbar interbody fusion (LLIF). We studied a retrospective cohort of 59 patients (34 males, 25 females; mean age, 68.9 years) who received single-level LLIF. Patients were classified into subsidence and no-subsidence groups. Cage subsidence was defined as any violation of either endplate, classified using radiographs and computed tomography (CT) images. After one year, we compared patient characteristics, surgical parameters, radiological findings, pain scores, and fusion status. We also compared the Hounsfield unit (HU) endplate value obtained on CT preoperatively. Twenty patients (33.9%) had radiographic evidence of interbody cage subsidence. There were significant differences between the subsidence and no-subsidence groups in sex, cage height, fusion rate, and average HU value of both endplates (*p* < 0.05). There were no significant differences in age, height, weight, or body mass index. Moreover, there were no significant differences in global alignment and Numerical Rating Scale change in low back pain, leg pain, and numbness. Despite suggestions that patients with lower HU values might develop cage subsidence, our results showed that cage subsidence after single-level LLIF was not associated with low back pain, leg pain, or numbness one year post-operation.

## 1. Introduction

Lateral lumbar interbody fusion (LLIF) via the lateral retroperitoneal approach has gained popularity and has been widely adopted to achieve interbody fusion with fewer complications [1,2,3]. It can be performed by two approaches, (1) extreme lateral interbody fusion [4], which accesses the intervertebral disc via transpsoas, and (2) oblique Lateral Interbody Fusion [5], which is accessed via the oblique corridor between the aorta and the psoas muscle.

LLIF with a minimally invasive transpsoas approach has been used for indirect decompression of spondylolisthesis and for spinal correction of adult spinal deformities. The LLIF provides the ability to release, reconstruct and fuse the spine while simultaneously providing indirect decompression of the neural elements through disc space distraction and spinal alignment. This approach usually does not encounter great abdominal vessels.

The operative time, blood loss, and tissue damage are reduced compared to posterior approaches. As with other minimally invasive approaches, the postoperative pain and the return to activities of daily living are faster [6,7,8].

However, these LLIF procedures can result in several perioperative complications, including nerve injury, vascular injury, or endplate injury [3,9,10,11]. Among these, endplate injury often occurs during endplate preparation and cage placement [12]. Once it occurs, it may result in cage subsidence to adjacent vertebral endplates, leading to the loss of segmental lordosis, foraminal height, and postoperative indirect decompression. Several reports have described postoperative cage subsidence in LLIF series, with an incidence of 10–22% [13,14,15]. While many studies have included patients at multiple levels and a single level, the etiology of cage subsidence should be considered only between single levels to determine accurate cage subsidence. Therefore, this study aimed to investigate the incidence of cage subsidence after single-level LLIF and to investigate its clinical features (Figure 1).

## 2. Materials and Methods

Before study initiation, this retrospective study was approved by Tokai University School of Medicine, the House Clinical Study Committee, and the Profit Reciprocity Committee. All methods were performed following the relevant guidelines (approval no. 21R163).

### 2.1. Included Patients

All patients who underwent LLIF for degenerative lumbar disc diseases (from L1-2 to L4-5) between April 2018 and September 2020 were reviewed, and the operative data in the medical records were investigated. Patients who underwent the LLIF procedure for a degenerative lumbar condition or adjacent disc disease at a single level were included. Patients with significant lumbar scoliosis, grade 2 spondylolisthesis, or lumbar fracture were excluded. Preoperative information for all patients was assessed using standard radiographs, MRI scans, and computed tomography (CT) scans. The spine surgeon recorded the location of stenosis based on an evaluation of the preoperative imaging studies. Patients underwent single-stage treatment with LLIF, followed by posterior percutaneous pedicle screw (PPS) fixation but without posterior decompression. Every patient with posterior instrumentation had PPSs inserted. Clinical data or surgical data, including age, sex, body mass index (BMI), operation time, intraoperative bleeding, and length of hospitalization stay, were reviewed in the medical records.

The patient demographic details are shown in Table 1.

A total of 59 patients were eligible for inclusion in this study. At surgery, the mean age was 68.9 ± 10.6 years (range 25–89 years), and 42.3% (25/59) were female. There were 46 (78.0%) patients over the age of 65. Tobacco use was found in 12 patients (20.3%), and four patients (6.8%) used corticosteroids. Diagnoses were lumbar canal stenosis/lumbar degenerative spondylolisthesis (*n* = 51), foraminal stenosis (*n* = 6), and lumbar disc herniation (*n* = 2). The most common segment was L4-5 (41 segments, 69.5%), followed by L3-4 (16 segments, 27.1%), and L2-3 (2 segments, 3.4%). The average operation time was 92.3 min. The average estimated blood loss was 62.8 mL, and the average length of hospital stay was 15.0 days. Regarding posterior fixation, there were 49 patients (83.1%) with bilateral fixation and 10 patients (16.9%) with unilateral fixation. The choice of unilateral or bilateral PPS fixation was based on randomization. Immediately after the surgery and before discharge, there were sensory deficits in 13 (22.0%) patients. The preoperative stair-climbing score was 10 points for 53 patients and 5 points for six patients. After LLIF, the scores were 10 points for 44 patients, 5 points for 14 patients, and 0 points for one patient. Thirteen patients (20.6%) lost one rank score immediately after the LLIF at the time of discharge, indicating motor weakness in these patients.

According to the cage subsidence pattern of each segment, 59 patients were classified into one of two groups: subsidence or no-subsidence. We compared patient background, pre- and postoperative alignment changes, and Numerical Rating Scale (NRS) scores between the two groups.

### 2.2. Surgical Technique

The basic procedure of our LLIF was performed according to the surgical technique described by Ozgur et al., and as explained in our previous papers [16,17,18,19]. Briefly, all patients underwent single-level LLIF through a single incision, mini-open direct visualizing approach. Patients were placed in true lateral positions, and a horizontal skin incision was made. A blunt incision was made until it reached the vertebral body. The cartilage endplate was removed using a Cobb elevator and curette when treating the endplate. Cage size trials were followed by additional disc curettage and rasping of the endplates. The surgeon determined the appropriate cage size by combining preoperative images and intraoperative cage template findings. All cages were lordosis cages and were 18 mm wide. The implant was placed on the apophyseal ring so that it straddled both sides. All LLIF segments were supplemented with unilateral or bilateral PPS fixation.

### 2.3. Radiological Assessment

We analyzed the imaging data before and after the LLIF. The standing anteroposterior and lateral radiographs were obtained for all patients preoperatively and postoperatively. The images were obtained with the patient in a free-standing posture with the fingers placed on the clavicles.

The spinopelvic parameter measurements included the following: sagittal vertical axis (SVA), lumbar lordosis (LL), thoracic kyphosis (TK) at T5-L1, sacral slope (SS), pelvic tilt (PT), and pelvic incidence (PI). The following parameters of each intervertebral disc were measured: segmental disc angle (SDA) on the sagittal plane in the neural position and anterior, posterior, and average disc height (ADH, PDH, and AvDH, respectively). DH was defined on sagittal sequences as the adjacent endplate distance between the intervertebral disc space anterior and posterior edges. AvDH was calculated as the average of the ADH and PDH values. As described previously, the interbody cage position was evaluated based on the midpoint locality relative to the midpoint of the inferior endplate [20].

The CT scans were performed preoperatively and immediately postoperatively to assess cage position and instrumentation and identify possible endplate injuries. Additional CT scans were obtained about one year postoperatively to assess fusion status and cage subsidence. Hounsfield units (HUs) were also measured on both the cranial (inferior endplate) and caudal (superior endplate) endplates of the treated level in preoperative CT images (Figure 2).

The top images show the midsagittal and axial planes of interest in a slice of a CT scan of the cranial endplate (inferior endplate). The bottom images show the HU values of the midsagittal and axial planes of interest on the caudal endplate (superior endplate).

The preoperative axial CT image layer used for HU measurement visualizes the largest area of cortical bone in the endplate region. The region of interest (ROI) was chosen manually to fit the shape of each structure. After establishing a consistent and appropriate ROI, the PACS system calculated the mean HU values. At the layer of CT where the structure of interest was present, the HU of the ROI was measured, and the average values were calculated. Endplate injuries and cage subsidence were categorized as caudal (superior endplate) and/or cranial (inferior endplate) and classified using radiographs and CT images according to Marchi classification [14]. Briefly, in this system, Grade 0 is a loss of 0 to 24% of postoperative DH, Grade 1 is 25 to 49%, Grade 2 is 50 to 74%, and Grade 3 is 75 to 100%. Subsidence was deemed early cage subsidence (ECS) if it was evident on radiographs and/or CT images during hospitalization and was, therefore, the result of an intraoperative endplate injury. On the other hand, if there was no evidence of endplate injury on radiographs and CT within two weeks after surgery, but subsidence was detected on subsequent radiographs and/or CT, it was deemed delayed cage subsidence (DCS).

Profiles such as the height and length of the inserted cage were recorded. In addition, we investigated the location of the cage. The cage placement was normalized to the distance between the anterior vertebral border of the inferior endplate, and the center of the cage was measured and normalized to the anteroposterior width of the inferior endplate [20,21]. The center of the cage was defined as the midpoint between the anterior and posterior radio markers of the cage. If the cage position was <50% displaced, the cage was positioned anteriorly. Fusion was determined by lumbar radiographic examination and/or CT scans one year after surgery. The evaluation of fusion was initially by CT scan. Patients who could not undergo a CT scan one year after surgery were evaluated using lumbar radiographic examination. The criteria for fusion status from the CT scan were the presence of a bony bridge in the sagittal and coronal reconstruction planes and its partial or complete connections to the lower and upper endplates. That is, our fusion criteria also included cases of partial fusion. In contrast, the criteria for fusion in lateral dynamic radiography was the presence of regional motion of <3° and intervertebral translation of <3 mm, without having had a revision or evidence of instrumentation loosening at one year after surgery. If any defects were in any position, the fusion status was classified as pseudarthrosis [22].

### 2.4. Clinical Assessment

Pain quantity in the low back and leg area using an NRS with scale graduation of 0–10 (0 = no pain, 10 = maximal pain imaginable) was obtained from patients preoperatively at one-year follow-up. The NRS scores were obtained for low back pain (NRS_LBP_), leg pain (LP; NRS_LP_), and leg numbness (LN; NRS_LN_). Improvements in symptoms were evaluated by the change in NRS (ΔNRS; 12 months postoperative NRS score—preoperative NRS score). We also collected information on postoperative complications, including transient psoas weakness and thigh pain or numbness at discharge if the patient subjectively commented on new events by the time of discharge.

### 2.5. Statistical Analysis

Statistical analyses were performed using IBM SPSS Statistics (version 23.0; IBM Corp., Armonk, NY, USA). All values are expressed as mean ± standard deviation. The Shapiro–Wilk test was used to confirm the normality of the data distribution. For the primary analysis, Student’s *t* test or the Mann–Whitney *U* test was used to compare the two groups. Student’s t-test was used to analyze normally distributed data and the Mann–Whitney *U* test for nonnormally distributed data. Comparisons between groups for categorical variables were assessed using the chi-squared test (Fisher). The significance of the obtained results was judged at the 5% level.

## 3. Results

Table 2 shows details of the cage subsidence group. ECS was observed in nine patients (15.3%), DCS was observed in 11 patients (18.6%), and total postoperative cage subsidence was identified in 20 patients (33.9%) one year postoperatively. Unilateral endplate injury, indicated by damage of either endplate in an intervertebral disc, was noted at 16 levels, and bilateral injury, indicated by damage of both endplates in an intervertebral disc, was noted at four levels. Injury at the endplate cranial and caudal to the disc was noted at nine and 15 levels, respectively.

At one year after surgery, Grade 1 subsidence was measured in 11 patients (55.0%), Grade 2 in five patients (25.0%), and Grade 3 in four patients (20.0%). Furthermore, the cage subsidence level was L2-3 in one patient, L3-4 in 10 patients, and L4-5 in nine patients.

Table 3 summarizes demographic and radiological factors between the two groups. There were no significant differences in age (*p* = 0.220), height (*p* = 0.145), body weight (*p* = 0.231), BMI (*p* = 0.447), tobacco use (*p* = 0.963), steroid use (*p* =0.075), operative times (*p* = 0.147), blood loss (*p* = 0.176), hospital stay (*p* = 0.879), or fixation type of PPS (*p* = 0.657) between the groups. There were no significant differences in cage height (*p* = 0.053), cage length (*p* = 0.114), cage material (*p* = 0.307) or cage position (*p* = 0.315) between the groups.

The HU values of all cases’ cranial endplate and caudal endplates were at 310.1 ± 67.1 HU and 277.2 ± 70.6 HU, respectively, which were higher in the cranial endplate (data not shown, *p* = 0.011). The mean HU value of the cranial endplate in the no-subsidence group was 325.0 ± 68.4 HU, whereas it was 281.2 ± 55.2 HU in the subsidence group. The HU value was low in the cage subsidence group (*p* = 0.016). Similar trends were seen in the caudal endplates (*p* = 0.012). The average HU value combined with the cranial and caudal levels was also significantly lower in the cage subsidence group (*p* = 0.004).

The incidence of new postoperative thigh pain and numbness, or motor weakness was similar between the groups. When evaluated based on our bone fusion rate criteria, there was a statistically significant difference in the fusion rate between the two groups (*p* < 0.001). The fusion rate at one year postoperatively was found in 92.3% (36/39) of patients in the no-subsidence group, compared with 55.0% (11/20) in the subsidence group. The fusion rate one year after surgery was 79.7% (47/59) in all cases.

Changes in ΔADH (*p* = 0.055), ΔPDH (*p* = 0.711) and ΔAvDH (*p* = 0.125) did not differ between the two groups. It appears that each DH increases postoperatively. SDA was significantly increased in patients without cage subsidence, but SDA did not change before and after surgery in patients with cage subsidence. Single-level LLIF did not significantly differ in SVA, LL, TK, PI, PT, and SS before and after surgery in either group (Table 4).

There was no statistically significant difference in each ΔNRS between the two groups preoperatively and one year postoperatively. The NRS score for each postoperative pain was significantly improved with or without cage subsidence (Table 5).

## 4. Discussion

Two types of cage subsidence have been reported. ECS develops acutely during surgical procedures, and ECS development may have different pathological mechanisms and risk factors compared with DCS. In contrast, DCS is thought to result from biological remodeling at the cage–bone interface in a chronic fashion [23]. In spinal fusion, including LLIF, several factors have been reported to cause cage subsidence. Older age, female sex, cage size, multilevel cases, and osteoporosis may be risk factors for cage subsidence [24,25,26,27]. A systematic review reported that patients with poorer bone quality, those older than 65 years, and women should be counseled about high risks of both types of cage subsidence [28]. Anecdotally, it is also said that the cage subsidence may differ depending on the supplementary instrumentation. In a meta-analysis comparing cage subsidence rates, the cage subsidence rate was 22.1% in the stand-alone group and 15.4% in the instrumented group [15]. Based on these data, elderly patients with poorer bone quality may have significant cage subsidence. Most of our cases were patients aged 65 years or older (78.0%). In our case, the cage subsidence rate (33.9%) may have been high due to the older age.

It cannot be denied that cage subsidence may affect the results of indirect decompression. Since the causes of cage subsidence are multifactorial, surgeons should pay attention to every aspect to optimize the outcome, including careful patient selection, improvement of bone health, and a meticulous intraoperative surgical technique. Potential patient-related and procedure-related factors that may affect indirect decompression have been reported [29,30]. During LLIF surgery, the surgeon only has control over approach choices, cage selection, and cage placement. Cage selection and cage placement can affect indirect decompression by LLIF [21,31]. Therefore, the surgeon may consider placing a larger cage in a narrow disc space for indirect decompression at LLIF. It has also been reported that the greater the postoperative increase in DH by LLIF, the greater the DH loss throughout early follow-up [32]. This means that greater cage height has been shown to increase the risk of cage subsidence. It has been recommended that the cage height should be <12 mm to prevent excessive mechanical stress on the endplates [33]. In our study, there were no patients where the cage height was ≥12 mm, but subsidence occurred even if the cage height was not ≥12 mm. For this reason, it is necessary to insert a suitable cage height for the patient. We reported that suitable cage placement was more critical than cage height to ensure indirect decompression using LLIF [21]. A suitable cage size must be placed in a suitable disc position to provide indirect decompression and prevent endplate injury and cage subsidence. Of course, the width of the cage can also affect the subsidence of the cage. Given that only 18 mm cages are available in Japan, the effect of cage width on cage subsidence is one of the limitations of LLIF [14].

Moreover, the material of the cage can also cause cage subsidence. Polyetheretherketone (PEEK) is a suitable implant for spinal fusion and has radiolucent properties so that fusion can be appropriately evaluated [34,35]. However, one of the disadvantages of PEEK cages is their relatively low osseointegration properties due to the biofilm layer around the surface of the cage [34,36]. Therefore, the layer must grow around the cage to attach to the bone. It has been reported that titanium cages have less cage subsidence. Most of the cages used in the present study were made of PEEK. The impact of each cage material on cage subsidence needs further evaluation.

The effects of osteoporosis make up a large proportion of patient factors that influence decompression. Kim et al. noted that endplate injury might be affected by cortical bone strength [12]. It has been reported that the endplate cranial to the disc was thicker and of greater density than the caudal disc. Based on this biomechanical property, cage subsidence could be expected to occur at the weak endplate.

Some groups have reported that HU values may serve as an assessment of osteoporosis, given the moderate correlation between HU values and osteoporosis [37,38,39]. HU values have also evaluated studies of LLIF cage subsidence [40,41]. It was shown that the decrease in the HU value of preoperative CT correlates with the severity of cage subsidence after LLIF. In our study, as in previous reports, endplate injury or cage subsidence was more common in the caudal endplate (62.5%). Moreover, in contrast to previous evaluations of HU values performed on the vertebral body, in this study, we evaluated HU values at the level of the endplate. Our data showed that the mean HU values differed significantly between the groups. From these data, preliminary assessment of endplate HU value may reduce postoperative cage subsidence. We recommend premeasurement of the HU value with preoperative CT when planning LLIF for patients with lumbar degenerative disease. Furthermore, preventing errors in the surgeon’s technique is necessary to prevent cage subsidence. Surgeons should proceed with caution in patients with unparallel endplate orientation due to deformity, as well as rough endplate treatment.

The effect of cage subsidence on clinical outcomes and fusion rate is controversial. Some groups report that these are not affected by either intraoperative endplate injury or late-onset settling at one year post-operation [13,42]. It was previously shown that the clinical correlation between subsidence and clinical outcomes primarily depends on the severity of the subsidence [33]. The present study showed postoperative improvement in pain scores for NRS_LBP_, NRS_LP_, and NRS_LN_ after LLIF surgery. In the comparison of cage subsidence, although the subsidence group demonstrated a worse fusion rate than that of the no-subsidence group at one year post-operation, there was no statistically significant difference between the two groups in ΔNRS, and the clinical symptoms were generally improved. Moreover, the cage subsidence found in our study was considered minor, and further surgical intervention was not needed. From these data, cage subsidence seems unlikely to affect postoperative pain in the short term (i.e., about one year after surgery). Cage subsidence, especially DCS, is affected by bone resorption and remodeling until rigid arthrodesis occurs. The lower fusion rate in the cage subsidence group may be due to this effect.

The retrospective nature of this study contributes to several limitations. Due to the limited cohort size, firm conclusions cannot be drawn. The follow-up period was also relatively short, although it was previously noted that 6–18 weeks were sufficient to access cage subsidence [43]. The factor of surgeon’s caution in osteoporotic patients could not be considered. We did not assess bone mineral density, which may have influenced the occurrence of cage subsidence in the selected elderly population due to osteoporosis. It cannot be denied that age may also affect the results. Finally, since many cages are made of PEEK material, the impact of new cage models, such as porous-coated titanium cages, on cage subsidence warrants further investigation.

## 5. Conclusions

We examined the cage subsidence at a single level of LLIF in this study. Cage subsidence was seen in 33.9% of patients, and more commonly at the caudal endplate of the disc. It was suggested that patients with lower HU values might develop cage subsidence. Therefore, surgeons should be cautious of an aggressive attempt to restore disc height with a tall cage for patients with poor bone density as it may lead to endplate injury and/or cage subsidence. Our results showed that cage subsidence was not associated with low back pain, leg pain, or numbness at one year postoperatively.

## Figures and Tables

**Figure 1 jcm-11-01374-f001:**
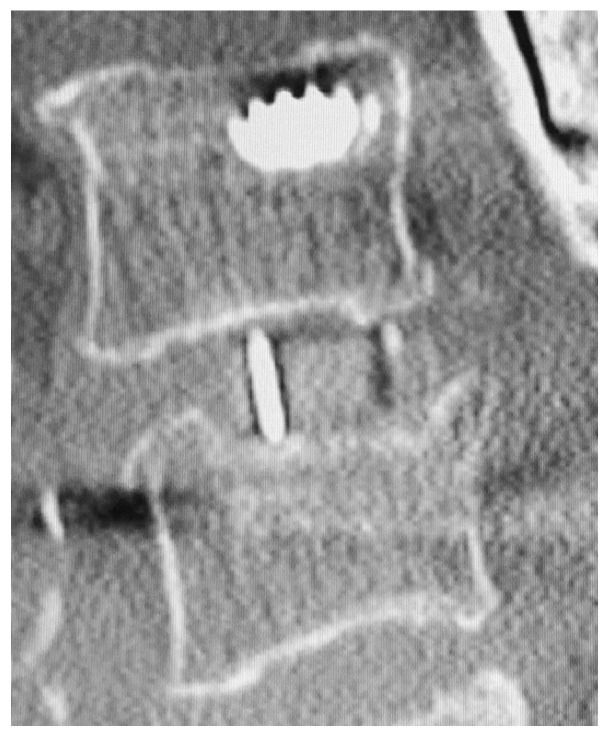
Cage subsidence examples. Polyetheretherketone cage single-level lateral lumbar interbody fusion (LLIF) postoperative computed tomography (CT) shows cage subsidence on the caudal endplate of the disc.

**Figure 2 jcm-11-01374-f002:**
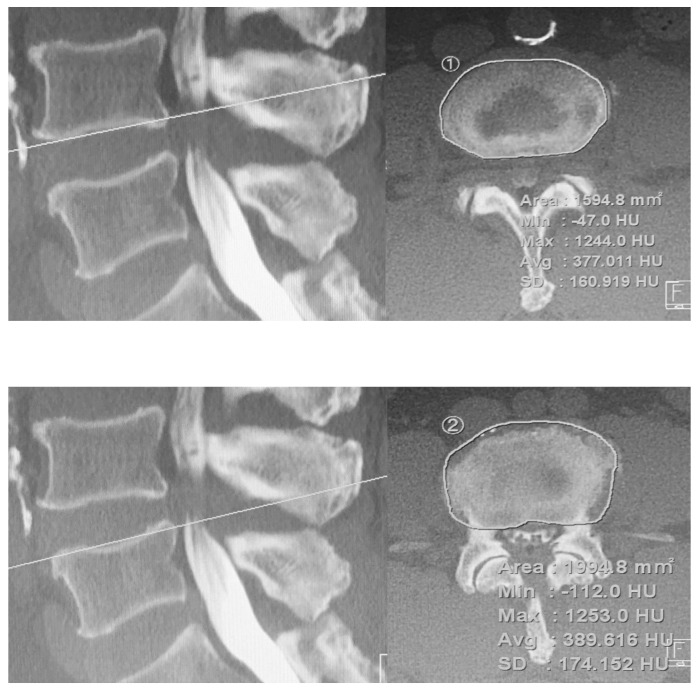
Computed tomography (CT) scans illustrate the method of determining the Hounsfield unit (HU) value with the use of an elliptical region of interest function.

**Table 1 jcm-11-01374-t001:** Patient demographic and treatment information in the present study. Data presented as mean (SD) or number of patients (%). BMI, body mass index; OR, operation; EBL, estimated blood loss; LCS, lumbar canal stenosis; LDS, lumbar degenerative spondylolisthesis; DLS, degenerative lumbar scoliosis; FS, foraminal stenosis; LDH, lumbar disc herniation; ASD, adjacent segment disease.

Characteristic		Data
No. of patients		59
Age (years)		68.9 (10.6)
≤65		13 (22.0)
>65		46 (78.0)
Sex (male/female)		34 (57.6)/25 (42.3)
Height (cm)		159.4 (9.9)
Body weight (kg)		61.6 (13.5)
BMI (kg/m^2^)		24.0 (4.1)
Tobacco use		12 (20.3)
Steroid use		4 (6.8)
Primary diagnosis	LCS + (LDS)	51 (86.4)
	FS	6 (10.2)
	LDH	2 (3.4)
Levels treated, *n* (%)	L1-L2	0 (0)
	L2-L3	2 (3.4)
	L3-L4	16 (27.1)
	L4-L5	41 (69.5)
	Overall	59
Average OR time (min)		92.3 (23.5)
Average EBL (mL)		62.8 (78.0)
Fixation type of PPS	Bilateral	49 (83.1)
	Unilateral	10 (16.9)
Average Length of stay (days)		15.0 (4.2)

**Table 2 jcm-11-01374-t002:** The numbers of segments with cage subsidence according to the location, types and disc level.

No.		Data
Early Cage Subsidence (ECS)		9/59 (15.3)
Delayed Cage Subsidence (DCS)		11/59 (18.6)
Cage Subsidence		20/59 (33.9)
By location	Unilateral endplate	16 (80.0)
	Bilateral endplate	4 (20.0)
	Endplate cranial to disc	9 (37.5)
	Endplate caudal to disc	15 (62.5)
Marchi Classification	Grade 1	11 (55.0)
	Grade 2	5 (25.0)
	Grade 3	4 (20.0)
Levels treated, *n* (%)	L1-L2	0 (0)
	L2-L3	1 (5.0)
	L3-L4	10 (50.0)
	L4-L5	9 (45.0)
	Overall	20

**Table 3 jcm-11-01374-t003:** Comparison of cage subsidence between two groups. Data presented as mean (SD) or number of patients (%). BMI, body mass index; OR, operation; EBL, estimated blood loss; PPS, percutaneous pedicle screw; * statistically significant; ^‡^ Comparison among groups.

Parameters		Subsidence (−)	Subsidence (+)	*p * ^‡^
No. of patients		39 (66.1)	20 (33.9)	
Age (years)		68.4 (12.0)	72.8 (6.2)	0.220
Sex (male/female)		28/11	8/12	0.019 *
Height (cm)		161.0 (9.2)	156.5 (10.9)	0.145
Body weight (kg)		63.1 (13.1)	58.6 (14.2)	0.231
BMI (kg/m^2^)		24.1 (3.5)	23.8 (5.0)	0.447
Tobacco use		8 (20.5)	4 (20.0)	0.963
Steroid use		1 (2.6)	3 (15.0)	0.075
Levels treated, *n* (%)	L1-L2	0 (0)	0 (0)	0.012 *
	L2-L3	1 (2.6)	1 (5.0)	
	L3-L4	6 (15.4)	10 (50.0)	
	L4-L5	32 (82.1)	9 (45.0)	
	Overall	39 (100)	20 (100)	
Cage height (mm)	8	8 (20.5)	1 (5.0)	0.053
	9	21 (53.8)	9 (45.0)	
	10	8 (20.5)	10 (50.0)	
	11	2 (5.1)	0 (0)	
	Ave	9.1 (0.8)	9.1 (0.6)	
Cage width (mm)	18	42 (100)	21 (100)	-
Cage length (mm)	45	2 (5.1)	1 (5.0)	0.114
	50	7 (17.9)	8 (40.0)	
	55	23 (59.0)	9 (45.0)	
	60	7 (17.9)	1 (5.0)	
	Ave	54.5 (3.8)	52.3 (3.8)	
Cage position (%)		44.8 (11.3)	49.2 (10.1)	0.315
Cage Material	PEEK	37	20	0.307
	Titanium	2	0	
Cranial endplate Hounsfield unit (HU)		325.0 (68.4)	281.2 (55.2)	0.016 *
Caudal endplate Hounsfield unit (HU)		293.5 (69.6)	245.4 (62.8)	0.012 *
Mean endplate Hounsfield unit (HU)		310.2 (56.5)	263.3 (54.0)	0.004 *
Average OR time (min)		95.4 (24.0)	86.4 (21.9)	0.147
Average EBL (mL)		59.2 (83.1)	69.8 (68.4)	0.176
Fixation type of PPS	Bilateral	33 (84.6)	16 (80.0)	0.657
	Unilateral	6 (15.4)	4 (20.0)	
Average Length of stay (days)		15.1 (4.6)	14.8 (3.5)	0.879
No. of transient motor weakness		10 (25.6)	3 (15.0)	0.355
No. of thigh pain and/or numbness		9 (23.1)	4 (20.0)	0.789
No. of Fusion rate at post-ope one year		36 (92.3)	11 (55.0)	0.001 *

**Table 4 jcm-11-01374-t004:** Preoperative, postoperative, and change from pre- to postoperative sagittal measurements. Data presented as mean (SD) ADH, anterior disc height; PDH, posterior disc height; AvDH, average disc height; SDA, segmental disc angle; SVA, sagittal vertical axis; LL, lumbar lordosis; TK, thoracic kyphosis; PI, pelvic incidence; PT, pelvic tilt; SS, sacral slope ^†^ comparison with pre op ^‡^ comparison between groups * statistically significant.

Radiological Parameter		Preoperative	Postoperative	ΔPost-Pre	*p* ^†^
ADH (mm)	Subsidence (−)	8.7 (4.1)	14.3 (2.4)	5.7 (3.3)	<0.001 *
	Subsidence (+)	8.9 (3.7)	12.6 (3.2)	3.7 (4.3)	0.002 *
	ALL	8.7 (4.0)	13.8 (2.8)	5.1 (3.7)	<0.001 *
	*p* ^‡^	0.803	0.031 *	0.055	
PDH (mm)	Subsidence (−)	5.5 (2.8)	9.1 (2.3)	3.7 (2.3)	<0.001 *
	Subsidence (+)	4.7 (2.3)	8.0 (2.5)	3.5 (2.0)	<0.001 *
	ALL	5.2 (2.6)	8.8 (2.4)	3.6 (2.2)	<0.001 *
	*p* ^‡^	0.242	0.102	0.711	
AvDH (mm)	Subsidence (−)	7.1 (2.9)	11.8 (1.8)	4.7 (2.4)	<0.001 *
	Subsidence (+)	6.8 (2.8)	10.4 (2.4)	3.6 (2.8)	<0.001 *
	ALL	7.0 (2.9)	11.3 (2.1)	4.3 (2.6)	<0.001 *
	*p* ^‡^	0.719	0.019*	0.125	
SDA (°)	Subsidence (−)	3.1 (5.7)	5.8 (3.9)	2.7 (3.8)	<0.001 *
	Subsidence (+)	4.2 (3.2)	5.7 (3.9)	1.4 (3.8)	0.144
	ALL	3.5 (5.1)	5.8 (3.8)	2.3 (4.0)	<0.001 *
	*p* ^‡^	0.446	0.923	0.223	
SVA (mm)	Subsidence (−)	68.6 (68.3)	59.4 (50.9)	−7.2 (59.0)	0.419
	Subsidence (+)	67.0 (50.8)	71.2 (46.8)	4.2 (41.0)	0.666
	ALL	68.0 (62.5)	64.0 (49.6)	−4.0 (54.0)	0.588
	*p* ^‡^	0.700	0.411	0.733	
LL (°)	Subsidence (−)	36.8 (17.4)	39.9 (12.2)	3.1 (11.5)	0.107
	Subsidence (+)	37.3 (13.8)	37.5 (16.6)	0.2 (10.9)	0.934
	ALL	36.9 (16.2)	39.1 (13.7)	2.2 (11.3)	0.158
	*p* ^‡^	1.000	0.542	0.200	
TK (°)	Subsidence (−)	23.4 (10.8)	25.7 (10.4)	2.1 (6.0)	0.025 *
	Subsidence (+)	24.4 (11.5)	24.4 (11.1)	0.0 (5.8)	0.987
	ALL	23.7 (10.9)	25.3 (10.5)	1.6 (6.1)	0.056
	*p* ^‡^	0.753	0.664	0.182	
PI (°)	Subsidence (−)	50.5 (8.4)	51.7 (8.0)	1.2 (4.5)	0.110
	Subsidence (+)	54.5 (8.3)	52.6 (7.4)	−1.9 (7.6)	0.303
	ALL	51.8 (8.5)	52.0 (7.8)	0.2 (5.9)	0.803
	*p* ^‡^	0.099	0.685	0.062	
PT (°)	Subsidence (−)	20.5 (7.6)	21.0 (7.0)	0.5 (4.9)	0.512
	Subsidence (+)	25.3 (8.2)	22.8 (8.1)	−2.5 (8.2)	0.217
	ALL	22.1 (8.0)	21.6 (7.4)	−0.5 (6.3)	0.591
	*p* ^‡^	0.037 *	0.408	0.270	
SS (°)	Subsidence (−)	30.0 (9.5)	30.6 (8.2)	0.7 (6.5)	0.522
	Subsidence (+)	29.2 (7.3)	29.8 (8.6)	0.6 (7.1)	0.734
	ALL	29.7 (8.8)	30.4 (8.3)	0.7 (6.7)	0.468
	*p* ^‡^	0.775	0.724	0.953	

**Table 5 jcm-11-01374-t005:** Each pain intensity between two groups. NRS_LBP_, numeric rating scale for low back pain; NRS_LP_, numeric rating scale for leg pain; NRS_LN_, numeric rating scale for leg numbness; ^†^ comparison between groups, ^‡^ comparison with pre-op, * statistically significant.

		Preope	Postope (12 M)	Change(Δ)	*p* ^‡^
NRS_LBP_	Subsidence (−)	6.6 (2.6)	2.7 (2.9)	−3.9 (3.3)	<0.001 *
	Subsidence (+)	5.6 (2.8)	3.4 (3.5)	−2.2 (4.4)	0.037 *
	ALL	6.2 (2.7)	2.9 (3.1)	−3.3 (3.8)	<0.001 *
	*p* ^†^	0.123	0.795	0.139	
NRS_LP_	Subsidence (−)	6.5 (2.9)	1.8 (2.2)	−4.8 (3.3)	<0.001 *
	Subsidence (+)	6.8 (2.6)	2.5 (3.3)	−4.3 (3.2)	<0.001 *
	ALL	6.6 (2.8)	2.0 (2.6)	−4.6 (3.2)	<0.001 *
	*p* ^†^	0.864	0.880	0.530	
NRS_LN_	Subsidence (−)	6.5 (2.9)	2.6 (2.7)	−3.8 (3.6)	<0.001 *
	Subsidence (+)	6.5 (3.5)	3.2 (3.5)	−3.4 (3.9)	0.001 *
	ALL	6.5 (3.1)	2.8 (3.0)	−3.7 (3.7)	<0.001 *
	*p* ^†^	0.593	0.935	0.645	

## Data Availability

The data presented in this study are available on request from the corresponding author. The data are not publicly available due to privacy or ethical restrictions.

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
