# Peer review of "Comparative Study of Cage Subsidence in Single-Level Lateral Lumbar Interbody Fusion"

_jcm, 2022, doi:10.3390/jcm11051374_

Round 1
Reviewer 1 Report
The manuscript “Comparative Study of Cage Subsidence in Single-Level Lateral 2 Lumbar Interbody Fusion” by Akihiko Hiyama, Daisuke Sakai,Hiroyuki Katoh, Satoshi Nomura, Masato Sato and Masahiko Watanab aimed to investigate the incidence of cage subsidence after single-level LLIF and to investigate its clinical features.
Below are my comments and remarks regarding the article:
1. lumbar scoliosis was exclude criterium and 2 patient had DLS, Degenerative lumbar scoliosis;
2. no control group
3. Has the implant been covered with bone grafts?
4. minimal endplate HU was -300? These are the values of the air
Author Response
We wish to express our appreciation to the reviewers for their insightful comments on our paper. The comments have helped us significantly improve the paper.
Following the reviewers' suggestions, we removed a few patients and re-evaluated the study.
Reviewers' comments:
Reviewer #1:
The manuscript “Comparative Study of Cage Subsidence in Single-Level Lateral 2 Lumbar Interbody Fusion” by Akihiko Hiyama, Daisuke Sakai, Hiroyuki Katoh, Satoshi Nomura, Masato Sato, and Masahiko Watanabe aimed to investigate the incidence of cage subsidence after single-level LLIF and to investigate its clinical features.
Below are my comments and remarks regarding the article:
lumbar scoliosis was excluded criterium and two patient had DLS, Degenerative lumbar scoliosis;
Response> We sincerely thank the reviewer for this kind, insightful comment. We agree with the reviewer's opinion, and we have revised the text to reflect your suggestion. The patients with DLS were deleted.
no control group
Response> We would like to thank you for your careful review of our work. We have already demonstrated that “According to the cage subsidence pattern of each segment, 59 patients were classified into one of two groups: subsidence or no-subsidence.” Therefore, the group without cage subsidence is the control.
Has the implant been covered with bone grafts?
Response> We sincerely thank the reviewer for this kind, insightful comment. In the PEEK cage, DBM and/or iliac bone were used in this study.
minimal endplate HU was -300? These are the values of the air
Response> Thanks for the reviewer's questions. Certainly, reviewers and readers will misunderstand this meaning. Thus, the min and max HU values calculated from PACS have been removed from the table.
We look forward to hearing from you.
Sincerely,
Akihiko Hiyama, MD, PhD
Department of Orthopaedic Surgery
Tokai University School of Medicine
Email: a.hiyama@tokai-u.jp
All the authors proudly approve the content presented in the manuscript and the fact that it has never been published or submitted for publication elsewhere.
Corresponding Author:
Akihiko Hiyama, MD, PhD
Department of Orthopaedic Surgery
Tokai University School of Medicine
Shimokasuya, Isehara
Kanagawa, 2591193, Japan Tel: 81-463-96-1121 (Ext.2320) Fax: 81-463-96-4434
Email: a.hiyama@tokai-u.jp
Reviewer 2 Report
The authors investigate the incidence and clinical features of cage subsidence after single level lateral lumbar interbody fusion. 21/ 63 patients (33%) had cage subsidence by 1 year postoperatively. There were no noted risk factors or adverse outcomes related to cage subsidence.
Overall, the authors do an excellent investigation of a clinical question that has already been well discussed in the literature. The lack of a unique topic is countered by a thorough investigation into the topic with a strong supporting discussion.
A major criticism of the paper is the fact that the patient population included patients with prior fusions at adjacent levels. This can confound the data, especially given that the authors specifically only investigated single level fusions to take away other confounding variables of multi-level lateral fusions.
Authors included pateints with adjacent level disease.
Author Response
We wish to express our appreciation to the reviewers for their insightful comments on our paper. The comments have helped us significantly improve the paper.
Following the reviewers' suggestions, we removed a few patients and re-evaluated the study.
Reviewers' comments:
Reviewer #2:
The authors investigate the incidence and clinical features of cage subsidence after single-level lateral lumbar interbody fusion. 21/ 63 patients (33%) had cage subsidence by 1 year postoperatively. There were no noted risk factors or adverse outcomes related to cage subsidence.
Overall, the authors do an excellent investigation of a clinical question that has already been well discussed in the literature. The lack of a unique topic is countered by a thorough investigation into the topic with a strong supporting discussion.
Response> We would like to thank you for your careful review of our work.
A major criticism of the paper is the fact that the patient population included patients with prior fusions at adjacent levels. This can confound the data, especially given that the authors specifically only investigated single-level fusions to take away other confounding variables of multi-level lateral fusions. Authors included patients with the adjacent level disease.
Response> Following advice, we have changed text and tables. Patients with the adjacent segmental disease were excluded, as pointed out by reviewer 2.
Section Editor: Agree with the suggestions above. Please amend before publishing.
Response> We genuinely hope that this manuscript will be reviewed educationally and be accepted for publication. Thank you for your consideration of my paper.
We look forward to hearing from you.
Sincerely,
Akihiko Hiyama, MD, PhD
Department of Orthopaedic Surgery
Tokai University School of Medicine
Email: a.hiyama@tokai-u.jp
All the authors proudly approve the content presented in the manuscript and the fact that it has never been published or submitted for publication elsewhere.
Corresponding Author:
Akihiko Hiyama, MD, PhD
Department of Orthopaedic Surgery
Tokai University School of Medicine
Shimokasuya, Isehara
Kanagawa, 2591193, Japan Tel: 81-463-96-1121 (Ext.2320) Fax: 81-463-96-4434
Email: a.hiyama@tokai-u.jp
Round 2
Reviewer 1 Report
I have no more comments